# Seasonal Changes in 25(OH)D Concentration in Young Soccer Players—Implication for Bone Resorption Markers and Physical Performance

**DOI:** 10.3390/ijerph18062932

**Published:** 2021-03-12

**Authors:** Joanna Jastrzębska, Maria Skalska, Łukasz Radzimiński, Agnieszka Niewiadomska, Artur Myśliwiec, Guillermo F. López-Sánchez, Michał Brzeziański, Thomas Rosemann, Beat Knechtle

**Affiliations:** 1Department of Pediatrics, Diabetology and Endocrinology, Gdansk Medical University, 80-210 Gdansk, Poland; mariajastrzebska@gumed.edu.pl; 2Department of Health and Natural Sciences, Gdansk University of Physical Education and Sport, 80-336 Gdansk, Poland; lukaszradziminski@wp.pl (Ł.R.); admysliwiec@wp.pl (A.M.); 3Centralny Ośrodek Medycyny Sportowej, 81-538 Gdynia, Poland; agnieszka.niewiadomska@coms.pl; 4Faculty of Health, Education, Medicine and Social Care, Anglia Ruskin University-Cambridge Campus, Cambridge CB1 1PT, UK; guillermo.lopez-sanchez@aru.ac.uk; 5Department of Endocrine Disorders and Bone Metabolism, Medical University of Lodz, 90-572 Lodz, Poland; brzezian1910@wp.pl; 6Institute of Primary Care, University Hospital Zurich, 8091 Zurich, Switzerland; thomas.rosemann@usz.ch (T.R.); beat.knechtle@hispeed.ch (B.K.); 7Medbase St. Gallen Am Vadianplatz, 9001 St. Gallen, Switzerland

**Keywords:** vitamin D, soccer, physical fitness, seasonal variation

## Abstract

Searching for potential relations between changes in 25(OH)D concentration and in physical fitness is an interesting scientific topic. Thus, the main purpose of this study was to investigate the changes in serum concentrations of 25(OH)D in young football players in conjunction with indicators determining bone resorption and physical fitness. A total of 35 young soccer players were tested during the autumn competition period. Biochemical analysis of blood, aerobic capacity, running speed and power tests (Wingate test, squat jump, counter movement jump) were carried out at the beginning and at the end of the competition period. Significant decrements in concentration of 25(OH)D from 33.9 ± 5.87 to 23.7 ± 5.58 ng/mL were noted after the analyzed competition period. There were no significant changes in aerobic capacity along the competition period. Significant improvements were observed in 10 m sprint, 30 m sprint (*p* < 0.001), maximal power (*p* = 0.011) and total work capacity (*p* = 0.024). We found that the effect of changes in the players’ physical fitness does not occur in relation to 25 OH(D) concentration but occurs when these changes are analyzed as a function of the duration of the observation period. Changes in physical fitness of soccer players are determined by other factors then 25(OH)D concentration.

## 1. Introduction

The multidirectional impact of vitamin D on the human body has been widely investigated. Vitamin D participates in immunological processes and shows anti-inflammatory, anti-apoptotic, antifibrinolytic and even anticancer effects [1].

The need to maintain an optimal level of 25(OH)D in athletes, including professional footballers and people with little physical activity during and immediately after winter in Northern European countries was suggested by Kopeć et al. [2], Solarz et al. [3] and Bezuglow et al. [4] in their studies. These authors diagnosed a significantly lower concentration of 25(OH)D in athletes at that time compared to the summer period. Despite the lack of a significant influence of low 25(OH)D concentration in winter on the level of bone resorption markers, they suggested a constant control of calcium, phosphorus and parathormone (PTH) concentration in blood. Dietary intake of vitamin D can be estimated from food tables and is usually between 100 and 400 IU daily. Maintaining an adequate level of vitamin D during winter months requires supplementation or a proper diet including such food products as oily fish. Aside this source of vitamin D content of most foods is between 50 and 200 IU per serving [5]. Therefore, unless a lot of fish with high vitamin D content is important diet component, it is unlikely to exceed 400 IU a day [6].

When reviewing the literature on the effects of vitamin D on human life functions, it can be seen that there are few reports of young, physically active people. The most frequent scientific papers have discussed the effects of vitamin D on skeletal muscle function [7,8,9], bone turnover [3] and the risk of injury [10]. Moreover, the frequent injuries in young athletes may result from a considerable deficiency of vitamin D related to exercise but also from a lack of exposure to sun light, especially in countries with the fewest daylight hours during winter [10,11,12].

With regard to the exercise capacity of young athletes, it can be assumed that vitamin D can have a moderate effect on improving aerobic performance and, to a greater extent, on anaerobic performance and muscle strength [9,13]. Dahlquist et al. [14], when examining athletes from different sports, found that there were different opinions on the positive effect of vitamin D on their aerobic performance. A study by Koundourakis et al. [15] confirmed that there was a correlation between the 25(OH)D concentration in professional footballers and the performance of the neuro-muscular system and the size of VO2max in the studied athletes, which could have a beneficial effect on the reduction in stress associated with intensive physical effort in athletes during the competition period. Significant correlations between the cardiopulmonary system efficiency as defined by the VO_2_max index and vitamin D concentration in 470 European adolescents aged 12.5–17.5 years (among others; Spaniards, Greeks, Belgians, Germans) were found by Valtuena et al. [16]. Other results were presented by Książek et al. [17].

The authors obtained no significant correlations between muscle strength as well as VO_2_max index in professional footballers examined in Poland during the winter; the highest vitamin D deficiency period in their bodies. Additionally, in Poland, Jastrzębska et al. [13] applied vitamin D supplementation in young footballers during an 8-week experiment in the winter preparation period. The authors concluded that the administration of vitamin D might have a significant impact on the improvement of aerobic fitness of players.

Similar correlations, both with regard to plasma concentration of 25(OH)D and vitamin D supplementation and anaerobic performance parameters, have been found by several authors in active people [18,19]. Hamilton et al. [20] found no correlation between the concentration of 25(OH)D and functional strength of skeletal muscles in professional footballers. On the other hand, Close et al. [21] found no significant and positive effects of vitamin D supplementation on the improvement of shoulder muscle strength in barbell press and leg muscle strength in the jumps and sprints of 20 m in athletes of various disciplines. Similarly, the same authors confirmed the significant effect of vitamin D supplementation on shortening the speed of running on the 10 m section and increasing the height of the hard jump in a group of professional footballers [22].

The performance period for youth football teams usually lasts from the beginning of September to the end of November or even mid-December, depending on the number of teams taking part in a competition. During this time, when applying proper training loads, the players should maintain a stable level of aerobic capacity, and the exercise capacity related to speed and explosive force may improve [23]. This is most often the result of training loads focused on short duration and high-power exercises [24]. The systematic control of players’ physical fitness is the basic task for coaches. The evaluation of components such as aerobic and anaerobic capacity, speed or power provide useful information about training efficiency.

There are few scientific papers on the seasonal variation of 25(OH)D concentration in football players in the same group of subjects [4,25]. Moreover, a lack of studies considering the training period and applied training loads in conjunction with 25(OH)D concentration is evident [26]. Taking into account the widespread vitamin D deficiency in young athletes, especially in Northern European countries, as well as the possibility of an increased vitamin D consumption during intensive training, we designed a study to show changes in serum concentrations of 25(OH)D in young well-performing football players during the autumn. The main purpose of this study was to show if these changes would affect the level of indicators determining bone resorption and physical fitness of the tested footballers during the performance period. It was hypothesized that the 25(OH)D concentration at the end of autumn will be significantly lower than at the beginning of performance period. Moreover, we assumed that players with higher 25(OH)D concentration at the beginning of the project will achieve a higher level of physical fitness at the end of competition period. Therefore, the 25(OH)D decrement may be an important factor determinant of maintaining or improving players’ physical fitness.

## 2. Materials and Methods

### 2.1. Design

The research covered a three-month period of sports training for junior footballers during the performance period, from mid-September 2019 (end of summer) to mid-December 2019 (end of autumn). During this time, the players should be optimally prepared for the league matches and maintain this level for a period of three months. Before the start of the research, it was assumed that the degree of insolation in the athletes’ place of residence from the beginning to the end of the research project would be reduced and would not affect the athletes’ vitamin D synthesis.

All the respondents performed the same football training program including exercises of special techniques and tactics as well as endurance, speed and muscle strength (Table 1). During the period of examination, it differed in subsequent microcycles, depending on the tactics of the game until the next match and the phases of deliberate increase or decrease in training loads, taking into account, among other things, the effects of post-workout fatigue of players. Before and immediately after the end of the testing project, the players were blood drawn to determine 25(OH)D and selected biochemical indicators. Measurements of aerobic and anaerobic physical performance, as well as the motion speed and explosive force were also conducted. In addition, the number of sunny days and the athletes’ diet during the three-month test period were documented. Statistical analysis of the results was carried out on the whole test group (TG) and with a division into groups. According to the median value of 25(OH)D concentration (32.3 ng/mL) obtained in the first test individuals were divided into a lower 25(OH)D concentration group (LDG, < median (Me)) and a higher 25(OH)D concentration group (HDG, > Me). Another group division was made according to the changes (drops) in concentration of 25(OH)D in blood serum completion (individual changes before and after the project). Players were split into a small drop group (SDG, drop < Me) and a high drop group (BDG, drop > Me).

A typical micro-training cycle implemented during the research project is shown in Table 1. In the morning, the players were carrying out team exercises of football techniques and tactics and upgrading their exercise capacity. In the afternoon, three days a week, the players trained individually in a group of goalkeepers, defenders, midfielders and forwards (formations). Thus, a typical weekly training load contained 6–8 training sessions and one league game (12–15 h a week). The training experience of the study participants was between 10 and 12 years.

### 2.2. Sample

A total of 40 young football players joined the research project. The respondents represented the top level in their age category. They took part in the Central Junior League games in Poland. Some of them (70%) lived in the school dormitory and ate in the same way every day. The rest lived at home conditions. All players attended a sports class with a football profile at the Private High School. The players’ diet was standard with more vegetables, fruit and dairy products (sports diet). One month before and during the experiment the athletes consumed no vitamin supplements and other sports supplements with vitamin D. In accordance with the guidelines of the Helsinki Declaration, the athletes and legal guardians of underage athletes were informed in detail about the testing procedures and gave written consent to participate in the testing. In addition, the athletes were informed of the possibility of withdrawing from the experiment at any stage. In addition, due to the low air temperatures in autumn, the athletes we studied wore full-body sportswear, so the lack of UV exposure was one of the factors reducing their natural vitamin D synthesis. Apart from UV rays, an important factor in obtaining vitamin D in humans is their daily diet. In accordance with the Helsinki Declaration of 2013 the Local Bioethics Medical Committee in Gdansk (Poland) approved the experiment and agreed to the study (KB consent number−26/19). The participants who fulfilled the inclusion criteria gave their written consent after being informed about the study objectives, procedures and methodology.

### 2.3. Inclusion and Exclusion Criteria

All the players were members of the same club and performed identical training loads. Only participants who completed at least 85% of training sessions were included in the study. A total of 5 players did not meet these criteria (2 players were injured, another 2 missed more than 15% of training sessions, and 1 player was excluded due to illness). Finally, 35 players were included in the analysis of the results (age: 17.5 ± 0.6 years, body mass 71.3 ± 6.9 kg, BMI 22.2 ± 1.8 kg/m^2^).

### 2.4. Procedures

#### 2.4.1. Degree of UV Radiation and Cloud Cover

The research was conducted in the northern part of Europe, in Poland (Gdynia, 54.50° N 18.55° E, 33 m above sea level). The degree of UV radiation and cloud cover during the project duration, i.e., between 15th September 2019 and 15th December 2019 was recorded on the basis of historical data from the weather online service for the city of Gdynia (Figure 1) [27].

#### 2.4.2. Calculation of 25(OH)D, Calcium (Ca), Phosphor (P) and Parathormone (PTH) Concentration in Blood Plasma

Biochemical analysis of blood was carried out in an accredited analytical laboratory (Diagnostics-medical laboratories, Gdynia, Poland). The Sysmex XE 2100 D and XT 4000 analyzers (Sysmex Europe GmBH, Norderstedt, Germany) were used in fluorescent flow cytometry technology. The serum concentration of 25(OH)D was calculated using Chemiluminescence (CMIA) (Liaison XL, DiaSorin, Saluggia, Italy) using 25OH Vitamin D Total Assay reagent. The intra-assay CV of the method was 2.4–6.4%, with respect to the range; 0–20 ng/mL deficit, >20–30 ng/mL suboptimal concentration, >30–50 ng/mL optimal concentration, >50–100 ng/mL high, >100 ng/mL potentially toxic, >200 ng/mL toxic [28]. Total calcium (Ca) in the blood serum was calculated using colorimetry, with Ca_2_ (Roche Cobas 6000, Roche Diagnostics, Mannheim, Germany). The intra-assay CV of the method was 2.5%, with respect to the range 8.4–10.2 mg/dL. Phosphorus in blood serum calculated by spectrophotometric method with PHOS2 reagent (Roche Cobas 6000). The intra-assay CV of the method was 1.4% with respect to the range 2.7–4.9 mg/dL. Parathormone (PTH) in blood plasma (EDTA) was calculated by chemiluminescence (CMIA), Elecsys PTH reagent (Roche Cobas 8000, Roche Diagnostics, Mannheim, Germany). Indirect precision was 3.1%. The intra-assay CV of the method was 3.1%, with respect to the range 15–65 pg/mL.

#### 2.4.3. Calculation of Average Vitamin D Intake

Diet (V.6.0) software developed by the Institute of Nutrition (Poland, 2018) and a vitamin D calculator in food products was used to calculate the daily vitamin D intake for each player during the first and last week (from Monday to Sunday) of the project.

#### 2.4.4. PWC_170_ Test

The Physical Work Capacity (PWC_170_) test was performed under laboratory conditions. The aim of this test was to determine the index of total work performed at the heart rate (HR) of 170 bpm using extrapolation, and to estimate VO2max indirectly. The subjects refrained from any physical training or activity at least one day prior to the test. Just before the test, body mass was measured with an electronic scale Tanita TBF 300M (Tanita, Tokyo, Japan). The PWC170 test was performed on a cycloergometer (Monark Ergomedic 830 E, Monark, Sweden). Before the test, each subject adjusted the saddle of the cycloergometer to his body height and had a portable heart rate monitor attached (Polar Electro OY, Kempele, Finland). During the test, the load was administered individually so that the first 5-min effort raised the player’s HR up to 120–130 bpm and the second up to 150–160 bpm. The mean value of the HR was registered in the last minute of each 5-min stage [29].

#### 2.4.5. Peak Power and Total Work Test

A 30-s Wingate test on a cycle ergometer (Monark Ergomedic 894 E, Monark, Sweden) was used to assess the peak power and total work. A relative load corresponding to 7.5% of the subject’s body mass was applied. Before performing the test, the participants completed a 10-min warm-up, including pedaling at a frequency of 60 rotations per minute (RPM), with a relative load of 1.2 W∙kg^−1^ and three rapid accelerations between the 7th and 10th min. After the warm-up, the subjects performed five minutes of stretching and relaxing exercises and then started the test [30].

#### 2.4.6. Sprint Test

Before the test, the players performed a 20-min warm-up involving two 10 m and one 30 m sprints. The sprint times were recorded by double photocells (Smart Speed electronic system, Fusion Sport, Cooper Plains, Australia) positioned at the starting (0 m) and finishing lines (10 and 30 m) at a height of 0.7 and 0.9 m. The subjects performed two maximal attempts for the 10 and 30 m distances. Only the best (the shortest) times were used in the subsequent analysis. Each participant started from a standing position, with his front leg on the starting line (0 m). The resting periods were 120 s after 10 m and 240 s after the 30 m sprint.

#### 2.4.7. Jump Test

Before the test, the students performed a 20-min warm-up involving five vertical jumps. The test comprised two maximal vertical jumps without (Squad Jump, SJ) and two with swing arms (Counter Movement Jump, CMJ). The resting period between jumps was two minutes. Only the best (the highest) jump was used in the subsequent analysis. Then, after a 5-min break, the athletes made 10 possibly highest jumps one after another without a rest break. The average value of the 10 jumps was calculated for performance analysis.

### 2.5. Statistical Analyses

Compliance of the distributions with normal distribution was verified using the Shapiro–Wilk test. In case of compliance with normal distribution, the significance of differences between PRE and POST measurements was calculated using the *t* test for dependent trials, while in case of non-compliance with normal distribution, the non-parametric equivalent of the Wilcoxon Paired Test was used. Following the grouping due to changes in serum 25(OH)D concentration (large drop and small drop Group) and 25(OH)D concentration in the first test (deficit and optimal group), a two-factorial variance analysis (ANOVA) was performed for repeated measurements. This analysis was used to assess both inter-group and intra-group effects. Moreover, the post hoc observed power was calculated. To determine the magnitude of the effect a partial eta square (*_p_*η^2^) was calculated. The ES was classified as small (≥0.01), medium (≥0.06) and large (≥0.14) [31]. The possible relations between the level of 25(OH)D concentration and the results of physical measurements were calculated using Pearson’s correlation and were fixed into following categories: very strong (r ≥ 0.80), moderately strong (r = 0.60–0.79), fair (r = 0.30–0.59) and poor (r ≤ 0.29). The significance of differences for all analyses was calculated at *p* < 0.05. The statistical analysis was performed using Statistica version 13.0 (TIBCO Software Inc., 2017 Palo Alto, CA, USA).

## 3. Results

At the end of summer (PRE), most of the footballers tested had an optimal serum concentration of 25(OH)D in the blood. No player showed a deficit value. After three months, at the end of autumn, significant changes in this indicator were recorded. The deficit was reported in 29% of the footballers, and no optimal values were found in the study group (Figure 2).

The level of 25(OH)D was significantly correlated with CMJ power (r = 0.35, *p* = 0.04) and average power in 10 jumps test (r = 0.36, *p* = 0.03). The relations with 25(OH)D concentration and other physical measurements (aerobic capacity, anaerobic capacity, speed) were insignificant.

In terms of blood rates and anaerobic performance of the footballers (*n* = 35), significant changes in performance were found at the end of autumn (POST). There were no statistically significant changes in the indicators characterizing the aerobic capacity of players (Table 2).

With regard to the 25(OH)D concentration level, both the LDG and HDG subgroups observed significant changes in the end of the project indicators except for PWC170, VO_2_max and SJ. High values of *_p_*η^2^ between groups and within groups were observed for 25(OH)D. The effect magnitude for PTH changes was on average. A significant reduction in the 10 and 30 m sprint and average power during the 10 jumps test was noted in LDG at the end of autumn. In the HDG group, on the other hand, significantly shorter running times of 10 and 30 m and higher maximum powers were recorded in the Wingate test (Table 3).

Significant changes in Ca and P were noted in both BDG and SDG at the end of the research project. Furthermore, a statistically significant lower level of PTH at the end of competition period was reported in SDG. Moreover, with the exception of PWC_170_, VO_2_max and SJ, the effect of PRE-POST changes in the indices characterizing the aerobic and anaerobic capacity of footballers was low or minor (Table 4).

## 4. Discussion

In view of the fact that the deficiency of vitamin D among the large populations of people living in Northern European countries is well known, we have designed a study in which we wanted to show that the concentration of this vitamin is variable at different times of the year. In addition, if these changes are significant, whether they can affect the bone resorption and exercise capacity of young footballers.

The main achievement in our work is that, based on statistical calculations, we have presented that between the end of summer and the end of autumn, the concentration of 25(OH)D in the blood of young football players decreases significantly. On the other hand, by analyzing the initial 25(OH)D concentration in the test subjects before the research project and its greater or lesser drop after the project, we found that the effect of changes in the physical fitness of players does not occur in relation to subgroups but occurs when these changes are analyzed as a function of the duration of observation. Seasonal changes in vitamin D concentrations are linked to its endogenous production and depend on many factors related to the time of day, time of year, latitude and the large amount of melanin in the skin, since it binds UV rays [32]. Additionally, clothing and sunscreens can effectively inhibit its production [33]. In the latitude where the study was carried out (Northern Europe), sun exposure in summer can last up to 9 h and only 3 h in autumn [34]. During our study period, UV radiation was from low to very low values (Figure 1).

On the basis of the analysis of the diet of the tested athletes living at home (30%), we found that their daily vitamin D intake was 165.35 ± 18.52 IU/day in the first week and 179.46 ± 14.42 in the last week of the research. Furthermore, daily vitamin D intake in players living in the boarding school (70%) in the first and last week of the study was 145.21 ± 18.52 and 155.46 ± 15.47, respectively. These values are many times lower than those recommended for the Polish youth population (800–2000 IU/day, i.e., 20–50 μg/day) in the period from September to April, where the percentage of people with vitamin D deficiency is at the level of 73.5–83.2% [35,36]. Brustad et al. [37] confirmed in their studies that a proper diet, due to vitamin D resources, can have a beneficial effect on improving vitamin D levels. In a study of 60 volunteers in Norway, they recorded that their average vitamin D concentration was higher at the end of summer and in December. The authors found that a diet rich in vitamin D (fish and fish fats), especially in winter, masked its seasonal changes and led to atypical changes during the year. Among the participants we examined, we did not find such a relationship, which could be associated with vitamin D deficiencies in their diet and the lack of its purposeful supplementation.

Our research has unequivocally shown that the decrease in 25(OH)D concentration in the athletes’ blood is significant in the autumn. The worrying fact is that the number of athletes with optimal values has dropped from 77% at the beginning of the fall to 14% at the end. Furthermore, as many as 29% of the athletes in the second period of testing had a deficit of 25(OH)D (Figure 2). None of the scientific papers available to us regarding the seasonal variations of 25(OH)D have analyzed the effect on the level of blood parameters and physical performance of the time decrease in 25(OH)D in athletes of the same team.

Analyzing the results of our tests for the whole group of athletes (*n* = 35) without taking into account the concentration of 25(OH)D and the dynamics of its decrease, we found that their anaerobic performance indicators improved significantly, and their aerobic performance was at the same level. In our opinion, this is in line with the training plan. Nobari et al. [24] confirmed, in accordance with our assumptions, that during the starting period, due to very high intensity of effort during championship matches and the use of high-intensity and low-volume exercises, an improvement in anaerobic capacity of the players with simultaneous stabilization of their aerobic endurance is observed (Table 2). Skalska et al. [26], while researching young football players during the preparation period (winter), found that the strength of the training load may be greater than the changes in blood concentration of 25(OH)D. Similar results were obtained with regard to the starting period for footballers in our study.

By studying the effect of the initial concentration of 25(OH)D on changes in the physical performance indicators of footballers during the start period, we found that the players with a lower value of this indicator (LDG) achieved a significant improvement in the sprint run on 10 and 30 m and after 10 jumps on a strain gauge mat. The HDG participants significantly improved in all performance indicators except total work (J/kg) and SJ (W/kg). In addition, the level of aerobic performance indicators did not change significantly in any of the groups. Similar conclusions were presented by Dahlquist et al. [14] after reviewing the literature on the influence of vitamin D on the level of indicators characterizing exercise capacity of individuals with different physical activity, age, gender and occupational activity. With regard to the indicators determining anaerobic capacity of the studied individuals, the authors quoted as well as Koundourakis et al. [15] and Wynon et al. [38] found high correlations with high levels of vitamin D concentration. Barker et al. [39], examining the relationships between high concentration of 25(OH)D in blood in adolescents and rapid muscle regeneration after intensive exercise, found high correlations. In turn, Ward et al. [40] suggested that people with a higher 25(OH)D level achieve a significant improvement in the height and motion speed. However, such correlations were not obtained by Bezuglow et al. [4] when studying young footballers during winter, in whom they observed no significant correlations between 25(OH)D concentration, running speed and muscle power. However, with regard to aerobic capacity, the results of other authors were inconclusive. For example, Fitzgerald et al. [41] did not observe changes in aerobic capacity in athletes with higher concentrations of 25(OH)D, whereas Forney et al. [42] and Jastrzębska et al. [13] observed higher values of VO_2_max in vitamin D supplements. Taking into account the research in our work, we found that the vitamin D resources that the athlete’s body is able to store after a period of abundant sunshine in the summer affect more effectively the level of his anaerobic exercise capacity. It is, however, an open question whether the effect of the changes we have observed relates more to the differences in the level of 25(OH)D in the athletes or the effect of the applied training loads. However, with the exception of 25(OH)D and phosphorus concentration, no effect of changes between LDG and HDG groups was found. As more statistically significant changes in anaerobic performance indicators were recorded during the statistical intra-group analysis, we can assume that higher concentrations of 25(OH)D in athletes at the beginning of the test period may have had a statistically insignificant positive effect on the improvement of anaerobic performance of athletes (Table 3).

Analyzing the results of the tests, taking into account the high and low drop of 25(OH)D in the blood of the athletes, it was found that this factor does not have a significant effect on intergroup and intragroup changes in their aerobic and anaerobic capacity. The two-factor ANOVA showed only significant values of *_p_*η^2^ for some physical fitness indicators as a function of time. Changes in 25(OH)D concentration in footballers can be considered interesting. A higher decrease was calculated for players with initially higher concentration, and the effect of these changes affected both the time function and the group. However, this did not affect the decrease in physical fitness of players after the start period. Therefore, it can be assumed that the initial level of 25(OH)D may compensate for the dynamic decrease and not contribute to the decrease in the exercise capacity of the individuals tested (Table 4).

Changes in the basic bone resorption markers such as Ca, P, PTH and in the level of 25(OH)D in blood plasma were reported in TG. Kopeć et al. [2] and Galan et al. [43], studying footballers, found significantly higher concentrations of 25(OH)D and calcium and a lower concentration of PTH after the summer period compared to the end of winter. In our study, we obtained similar results after the summer period, but at the end of autumn, we calculated significantly higher concentrations of calcium and phosphorus and significantly lower concentrations of PTH. The same relationships apply both to the whole group of athletes and to the subgroups that take into account the concentration of 25(OH)D and its decrease over time. In terms of bone resorption, this is a beneficial metabolic effect. Excessive parathormone secretion has many consequences, as this hormone is considered to be a uremic toxin and may have an adverse effect on mineral management and bone metabolism [44]. According to Maïmoun and Sultan [45], high physical activity affects the level of bone resorption markers and depends on the type of exercise. However, Lombardi et al. [46] found no significant correlation between the level of 25(OH)D and bone resorption hormones, and their concentration changes were more related to training load than to changes in 25(OH)D in blood plasma. Similarly to our study, Kopeć et al. [2] registered a slight tendency to decrease the concentration of bone resorption markers in Polish footballers during winter. However, among our players, no simultaneous decrease in calcium and phosphorus in blood was observed, which can be considered a beneficial metabolic effect.

Authors are aware of some limitations of current research. Larger number of participants would increase the power of statistical analysis. However, usually soccer teams involve no more than 25 players. Therefore, 35 players included in this research seemed to be acceptable. Individual diet of some players is another limitation of our study. Authors made some efforts to exclude this disturbance (calculation of daily vitamin D intake); however, using an equal diet for all the players is recommended for future research.

## 5. Conclusions

Taking into account the results of our research, the hypothesis assuming that the concentration of 25(OH)D in football players will significantly decrease at the end of the analyzed competition period was confirmed. At the same time, i.e., during the performance period, the indicators characterizing the aerobic capacity of the players did not change and the anaerobic capacity improved significantly. Significant changes in bone resorption markers were observed independently from the 25(OH)D concentration. The results of this research suggest that supplementation of vitamin D in the Northern European countries during autumn is recommended to avoid a significant decrement of 25(OH)D concentration. However, these drops are not related with changes in physical fitness.

## Figures and Tables

**Figure 1 ijerph-18-02932-f001:**
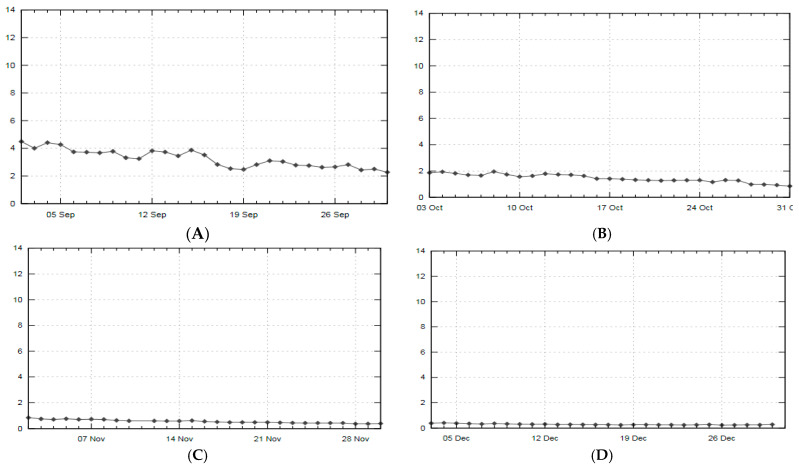
Scope of the UV index for the city of Gdynia (Poland) in the research period ((**A**)-September, (**B**)-October. (**C**)-November, (**D**)-December) [27].

**Figure 2 ijerph-18-02932-f002:**
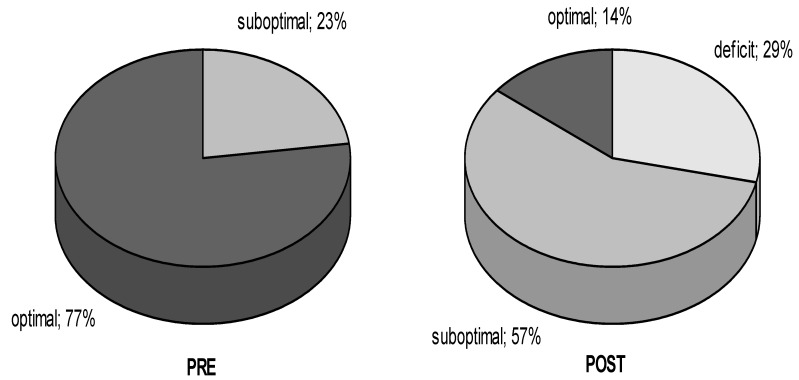
Percentage distribution of the number of footballers with different concentrations of 25(OH)D in blood serum at the end of summer (PRE) and at the end of autumn (POST).

**Table 1 ijerph-18-02932-t001:** Overview of the typical weekly training load during experiment.

Day of the Week	Training Drills	
	Morning	Afternoon
Monday	endurance, technical, tactical	free
Tuesday	speed, technical, small-sided games	individual training (formation)
Wednesday	stretching, regeneration	individual training (formation)
Thursday	plyometric and speed, tactical	individual training (formation)
Friday	coordination, tactical	free
Saturday	competition game	
Sunday	free day	

**Table 2 ijerph-18-02932-t002:** Changes in selected blood biochemical and physical performance indicators in the whole group (TG) of footballers (*n* = 35) before (PRE) and after (POST) testing program.

Variable	PRE	POST	*p*-Value
25(OH)D (ng/mL)	33.9 ± 5.87	23.7 ± 5.58 *	0.0000
Ca (mg/dL)	9.51 ± 0.28	9.98 ± 0.24 *	0.0000
P (mg/dL)	3.82 ± 0.49	4.35 ± 0.53 *	0.0000
PTH (pg/mL)	37.9 ± 14.90	32.9 ± 16.72 *	0.0070
PWC170 (kgm/min)	18.9 ± 2.15	18.5 ± 2.54	0.6465
VO_2_max (mL/kg/min)	49.9 ± 4.19	49.2 ± 4.62	0.5230
10 m (s)	1.75 ± 0.05	1.72 ± 0.05 *	0.0007
30 m (s)	4.24 ± 0.13	4.19 ± 0.15 *	0.0000
Total work (J/kg)	272.3 ± 15.02	275.4 ± 14.68 *	0.0239
P max (W/kg)	11.3 ± 0.80	11.6 ± 0.72 *	0.0106
SJ (W/kg)	48.3 ± 3.38	48.2 ± 3.75	0.6978
CMJ (W/kg)	55.3 ± 3.19	54.1 ± 4.24 *	0.0114
10 jumps (W/kg)	51.0 ± 3.15	49.9 ± 3.23 *	0.0035

Significant differences at: * *p* < 0.05. Ca—calcium, P—phosphorus, PTH—parathormone, PWC170—Physical Work Capacity, P max—maximal power, SJ—squat jump, CMJ—countermovement jump.

**Table 3 ijerph-18-02932-t003:** Changes in selected blood biochemical and physical performance indicators in young footballers in the group with low 25(OH)D (LDG) concentration (*n* = 17) and in the group with optimal 25(OH)D HDG concentration (*n* = 18) before (PRE) and after (POST) testing program.

Variable	LDG Group (*n* = 17)		HDG Group (*n* = 18)		*_p_*η^2^	OP
	PRE	POST	PRE	POST		
25(OH)D (ng/mL)	29.50 ± 3.52	20.76 ± 5.29 ***	38.13 ± 4.35	26.55 ± 4.31 ***	0.51 0.81	0.99 1.00
Ca (mg/dL)	9.44 ± 0.31	9.98 ± 0.25 ***	9.58 ± 0.24	9.97 ± 0.24 ***	0.77	1.00
P (mg/dL)	3.93 ± 0.58	4.54 ± 0.53 ***	3.72 ± 0.36	4.17 ± 0.48 **	0.12 0.48	0.55 0.99
PTH (ng/mL)	39.57 ± 13.25	32.85 ± 17.84 *	36.42 ± 16.54	33.03 ± 16.11	0.13	0.57
PWC_170_ (kgm/kg/min)	19.21 ± 2.16	18.90 ± 2.78	18.53 ± 2.15	18.22 ± 2.33	-	
VO_2_max (mL/kg/min)	50.80 ± 4.42	50.19 ± 5.06	49.05 ± 3.88	48.36 ± 4.11	-	
10 m (s)	1.75 ± 0.05	1.73 ± 0.04 *	1.74 ± 0.05	1.72 ± 0.05 *	0.29	0.95
30 m (s)	4.24 ± 0.10	4.19 ± 0.15 **	4.23 ± 0.15	4.19 ± 0.15 **	0.43	0.99
Total work (J/kg)	274.2 ± 16.61	276.5 ± 13.55	270.5 ± 13.59	274.3 ± 15.99	0.14	0.62
P max (W/kg)	11.48 ± 0.79	11.72 ± 0.52	11.14 ± 0.79	11.48 ± 0.87 *	0.23	0.86
SJ (W/kg)	49.28 ± 3.59	48.83 ± 4.44	47.39 ± 2.98	47.62 ± 2.96	-	
CMJ (W/kg)	55.53 ± 2.94	54.79 ± 4.30	55.12 ± 3.47	53.42 ± 4.20 *	0.18	0.73
10 jumps (W/kg)	51.36 ± 2.42	50.44 ± 2.86 *	50.66 ± 3.75	49.32 ± 3.53 *	0.22	0.85

Significant PRE–POST differences at: * *p* < 0.05, ** *p* < 0.01, *** *p* < 0.001; *_p_*η^2^—effect size, OP—observed power, Ca—calcium, P—phosphorus, PTH—parathormone, PWC_170_—Physical Work Capacity, P max—maximal power, SJ—squat jump, CMJ—countermovement jump.

**Table 4 ijerph-18-02932-t004:** Changes in selected biochemical indicators of blood and physical fitness in young football players in the group with large (BDG) and small (SDG) drops of 25(OH)D before (PRE) and after (POST) testing program.

Variable	BDG Group (*n* = 18)		SDG Group (*n* = 17)		*_p_*η^2^	OP
	PRE	POST	PRE	POST		
25(OH)D (ng/mL)	36.19 ± 6.14	22.0 ± 5.08 ***	31.3 ± 4.33	25.8 ± 5.59 ***	0.93 0.72	1.00 1.00
Ca (mg/dL)	9.52 ± 0.31	10.0 ± 0.25 ***	9.50 ± 0.25	10.0 ± 0.24 ***	0.76	1.00
P (mg/dL)	3.85 ± 0.44	4.4 ± 0.55 **	3.79 ± 0.55	4.3 ± 0.50 *	0.47	0.99
PTH (ng/mL)	38.28 ± 18.07	33.9 ± 16.69	37.56 ± 10.54	31.9 ± 17.23 *	0.13	0.56
PWC_170_ (kgm/kg/min)	18.92 ± 2.39	18.2 ± 2.51	18.80 ± 1.89	19.0 ± 2.60	-	
VO_2_max (mL/kg/min)	49.80 ± 4.53	48.4 ± 4.39	50.02 ± 3.89	50.2 ± 4.84	-	
10 m	1.74 ± 0.04	1.72 ± 0.04	1.75 ± 0.05	1.73 ± 0.05	0.29	0.95
30 m	4.23 ± 0.13	4.18 ± 0.15	4.24 ± 0.13	4.2 ± 0.15	0.42	0.99
Total work (J/kg)	272.1 ± 14.82	274.2 ± 15.07	272.51 ± 15.74	276.7 ± 14.58	0.15	0.65
P max (W/kg)	11.30 ± 0.82	11.6 ± 0.78	11.31 ± 0.80	11.6 ± 0.67	0.23	0.86
SJ (W/kg)	48.40 ± 3.99	48.5 ± 4.04	48.20 ± 0.60	47.8 ± 3.46	-	
CMJ (W/kg)	55.62 ± 3.54	54.0 ± 4.07	54.96 ± 2.78	54.2 ± 4.57	0.17	0.71
10 jumps (W/kg)	51.40 ± 2.97	49.6 ± 3.41	50.54 ± 3.38	50.2 ± 3.08	0.22	0.85

Significant PRE–POST differences at: * *p* < 0.05, ** *p* < 0.01, *** *p* < 0.001; *_p_*η^2^—effect size, OP—observed power, Ca—calcium, P—phosphorus, PTH—parathormone, PWC_170_—Physical Work Capacity, P max—maximal power, SJ—squat jump, CMJ—countermovement jump.

## Data Availability

The data presented in this study are available on request from the corresponding author.

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
