# Peer review of "Seasonal Changes in 25(OH)D Concentration in Young Soccer Players—Implication for Bone Resorption Markers and Physical Performance"

_ijerph, 2021, doi:10.3390/ijerph18062932_

Round 1
Reviewer 1 Report
authors addressed all concerns
Reviewer 2 Report
After the different reviews, authors have improved the manuscript and this can be published.
This manuscript is a resubmission of an earlier submission. The following is a list of the peer review reports and author responses from that submission.
Round 1
Reviewer 1 Report
The authors have improved the manuscript by incorporating the reviewer's suggestions. Just one consideration in Material and methods section:
- Ethical approval subsection must go within Sample subsection, for example: In accordance with the Helsinki Declaration of 2013 and the Local Bioethics Committee of the District Chamber of Physicians and Dentists in 115 Gdansk (Poland) approved the experiment and agreed to the study (KB consent number 116 - 26/19). The participants who fulfilled the inclusion criteria gave their written consent after being informed about the objectives, the procedures and the methodology of the study.
Reviewer 2 Report
The authors corrected many problems. Power calculations should be performed - even post hoc. Correlations between serum 25 (OH)D and performance measures should be included.
p. 9 It is not appropriate to analyze changes as a function of the duration of observation when only one time period (duration) was tested.